# A Framework for Open-Pit Mine Production Scheduling under Semi-Mobile In-Pit Crushing and Conveying Systems with the High-Angle Conveyor

**Dingbang Liu *** and **Yashar Pourrahimian**

School of Mining and Petroleum Engineering, University of Alberta, Edmonton, AB T6G 1H9, Canada; yashar.pourrahimian@ualberta.ca
* Correspondence: dingbang@ualberta.ca

**Abstract:** In-pit crushing and conveying (IPCC) systems have drawn attention to the modern mining industry due to the numerous benefits than conventional truck-and-shovel systems. However, the implementation of the IPCC system can reduce mining flexibility and introduce additional mining sequence requirements. This paper investigates the long-term production scheduling and the crusher relocation plan of open-pit mines using a semi-mobile IPCC system and high-angle conveyor. A series of candidate high-angle conveyor locations is generated around the pit limit, with a crusher located along each conveyor line. Each conveyor location is solved independently by an integer linear programming model for making production scheduling and crushing station decisions, aiming to maximize the net present value (NPV) considering the material handling and crushing station relocation costs. The production schedule with the highest NPV and the associated conveyor and crusher location is considered the optimum or near-optimum solution.

**Keywords:** in-pit crusher and conveyor; production scheduling; crusher location; conveyor location; integer linear programming; high-angle conveyor; hierarchical clustering





## 1. Introduction

The mining industry is advancing towards a more efficient, less energy-consuming, labor-intensive, and reliable process. In open-pit mining, material haulage costs constitute over 50% of the total mining cost [1]. Developing continuous mining extraction and transportation systems has become a promising direction for improving productivity and reducing operating costs. Conventionally, the truck-and-shovel system is used in open-pit mining to transfer materials out of the pit. These loading and hauling operations are adopted by over 80% of greenfield open-pit and open-cast mines [2]. The truck-and-shovel system is economical at the beginning of mine life. However, when the working surface goes deeper or broader, the truck-and-shovel system's costs rise exponentially. The pit expansion can result in a higher stripping ratio and waste level growth. In addition, the hauling distance and the elevation difference from the loading point to the destination increase sharply. The additional costs include more trucks needed in the fleet, more fuel consumption, tires and labor expenses, and more extended hauling road construction and maintenance [3,4].

In-pit crushing and conveying (IPCC) systems as alternatives to the truck-and-shovel system have attracted more attention today. The IPCC system is composed of a series of feeding, crushing, conveying, and discharging modules [5]. The belt conveyor haulage has much lower operating costs and are more energy-efficient than truck haulage, especially in horizontal-developed and large-scale open-pit mines [6].

Based on the equipment mobility, IPCC systems can be categorized into three types: fully-mobile, semi-mobile, and fixed systems [7]. The semi-mobile IPCC systems are the most popular category, as they can be easily transited from the truck-and-shovel system.

The crusher's relocation nature and the relatively lower initial investment make these types of IPCC systems more appealing to be adopted in modern mining activities [1].

On the other hand, semi-mobile IPCC systems have different strategic mine planning and sequence requirements than the truck-and-shovel counterpart; the crusher and conveyor cannot be relocated frequently due to the high relocation costs [8]. Moreover, the massive initial investment associated with IPCC systems forces the mine planners to design the mining strategy precisely in the long run, especially in large-scale mines [9].

For the conventional conveyor belt, the maximum inclination angle is generally determined by the repose angle of loose materials, which vary from 15° to 22° with respective angles of repose from 29° to 44° [10]. If the inclination angle exceeds the requirement, the material on the belt will slide back, as the internal friction of the material or the friction between the material and the belt surface is less than the material's gravity along the belt. However, this inclination angle is considerably lower than the pit slope (around 40°), which means the conventional conveyor cannot be directly implemented along the pit limit. Several solutions have been proposed to address this issue: conveying on a slot or through a tunnel, using existing haul roads, or applying high-angle conveyors (HAC). Among these options, a slot or tunnel requires additional waste to be excavated. The construction is subject to geological conditions; the existing haul road is several times longer than the straight conveyor route as the low ramp slope brings a high conveyor operating cost. However, applying a HAC can avoid many obstacles rising by the conventional counterpart. Because HAC can transport material at a deep angle along the pit wall, the conveyor belt length is minimized, and no extra waste material needs to be mined. Some types of HACs, such as the sandwich belt conveyor, are adaptable to multi-module sections using self-contained units [10]. So HACs can be extended to other levels during the mine development. For conveyor stability, especially in the deep open-pit mine, the HACs are usually anchored to the pit wall and mounted on concrete structures, which means the relocation of this system is rare [1,11]. This study considers the situation that the conveyor belt is fixed along the pit wall during the mine life. In this sense, two research problems arise based on the semi-mobile IPCC systems' configuration: (i) the production scheduling plan that gives the maximum net present value (NPV) with additional mining sequence and pit expansion restrictions and (ii) the crusher location-relocation plan that minimizes the material handling and crusher relocation costs. This paper combines the two problems into one model and simultaneously optimizes the production scheduling and crusher location problems. Furthermore, as different conveyor layouts can result in various mining sequences, a series of candidate conveyor locations around the ultimate pit limit (UPL) are investigated and their NPVs are compared.

This study establishes an integer linear programming (ILP) model that can simultaneously solve the production scheduling and crusher location problem to maximize NPV. This model is applied to different candidate HAC locations along the pit wall and finds the optimum scenario. The conveyor location that gives the overall highest NPV is considered as the optimum conveyor location, and the associated production scheduling and crusher relocation plan are also solved.

The following section reviews the available optimization models for IPCC systems. Section 3 describes a methodology to determine a series of candidate conveyor lines around the pit limit and a clustering technique to reduce the number of mining units. Section 4 presents the mathematical programming formulation. Section 5 presents a case study and discusses the implementation results, and the paper concludes in Section 6.

## 2. Literature Review

Many studies have been focused on crusher location and relocation plans in IPCC systems to minimize the truck's transportation costs. Some earlier works adopted the discrete event simulation method to solve the problem. Sturgul [12] simulated an in-pit crusher location using the GPSS (general-purpose simulation system) language. The author simulated the truck-and-shovel transportation time for each possible crusher location deter-

mined the situation with the minimum cycle time. Peng and Zhang [13] did similar work. They compared a variety of possible crusher locations and selected the one that generated the highest mining capacity. More recently, Konak et al. [14] established a trial-and-error process to enumerated different possible crusher locations on a level basis. The level gives the minimum average haulage distance as the optimum location. They also introduced an upward haulage coefficient to adjust hauling distances from different levels. Some recent works applied facility location problems to find the optimum crusher locations in the long-term plan horizon. Rahmanpour et al. [15] considered each of the possible crusher locations as a hub node. They solved the problem by an integer-programming model to minimize the total transportation cost (truck haulage cost and crusher relocation cost). Paricheh et al. [16] considered the crusher location/relocation as a time-dependent dynamic facility location problem. They examined the haulage and crusher relocation costs for each subsequent year from payback time as the IPCC system installation timing. They solved each case independently to find the minimal costs scenario. Paricheh et al. [2] develop a heuristic framework based on the dynamic location problem to solve the transition time from a pure-truck system to an IPCC system. They combined the two integer linear programming models based on a heuristic approach to obtain the IPCC application's optimum timing and corresponding crusher locations. The problem optimizes NPV in consideration of transportation costs. Abbaspour et al. [17] solved the crusher relocation plan by the transportation problem. They defined each mining unit as a source and each pit level where crushers can be located as a destination. Then, they investigated different crusher relocation intervals and defined the case with the lowest operating and relocation costs as the optimum plan.

Some studies have considered IPCC systems optimization as a production scheduling problem. Nehring et al. [8] compared the NPVs of different mining sequences of the pure truck, semi-mobile, and fully-mobile IPCC systems. They found that the IPCC system is more applicable for large-scale mines with large horizontal extensions and stable mining plans. Hay et al. [18] investigated the effect of semi-mobile IPCC systems' implementation on the pit limit. First, they determined the conveyor wall's optimum orientation, which gives maximum present values. The UPL was generated by the network flow method with the additional conveyor wall requirements. Jimenez Builes [19] used a mixed-integer goal programming model to maximize the NPV, and instead of a fixed production rate, a set of goal deviational variables and penalties were set. Paricheh and Osanloo [20] proposed an integrated mixed-integer linear programming model to solve semi-mobile IPCC system planning problems synchronously. The model comprises three parts: open-pit mine production scheduling, crusher relocation planning, and truck fleet sizing/replacement planning. Although they specified a set of initial candidate conveyor locations, they did not consider the conveyor wall location. Samavati et al. [21] solved the mine production scheduling problem under fully-mobile IPCC systems, with additional sequence constraints.

Although many studies have investigated IPCC systems' operating and capital costs, a few have analyzed the production scheduling problem under these systems. Furthermore, most recent studies focus on crusher location and relocation plans based on a set of predetermined candidate sites (e.g., the centroid of each level). However, finding the proper candidate locations is a critical challenge in real cases [20], and a greater number of these locations can significantly increase the complexity of the model. Additionally, no research has been found that considered the production scheduling and in-pit crusher location problems simultaneously; both are the key factors in the IPCC system optimization. This study solves the scheduling problem from the conveyor location's perspective, aiming to propose a new mathematical framework for optimizing both HAC's and crusher's locations under semi-mobile IPCC systems that maximize the NPV while considering the material handling and crushing station relocation costs.

## 3. Methodology

*3.1. Theoretical Framework*

This paper presents a situation that the HAC is anchored along one side of the pit wall throughout the mine life. The excavated material should be trucked towards this side of the pit, and the candidate crusher locations are on the conveyor line. This implementation introduces additional mining direction requirements; mining starts from the conveyor side and then expands to the other side of the level.

Figure 1 shows that the extraction of blocks closer to the conveyor has precedence over others.

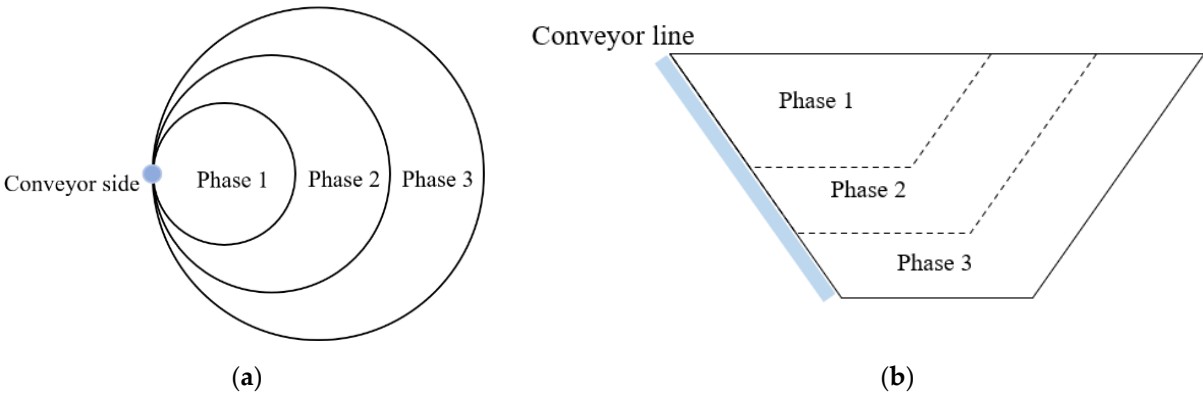

**Figure 1.** Schematic view of the mining method with the high-angle conveyors (HAC) system along one side of the ultimate pit limit (UPL): (**a**) plan view and (**b**) vertical view.

Because the belt conveying is a more efficient elevating method over trucking, this study assumes the trucks only transport materials from the working face to the crusher. The semi-mobile IPCC systems maintain some flexibility due to the trucking portion; thus, the crusher's allocation along a specific conveyor line cannot significantly affect the production scheduling and the economic comparison of different conveyor schemes. The mining sequence only changes subject to the various conveyor layout and results in different NPVs. The developed model finds the optimum conveyor location that gives the maximum NPV.

Figure 2 illustrates the framework of the methodology used in this paper. Starting from a known UPL, a series of candidate HAC locations are selected. In selecting the candidate locations for HAC, the pit wall's geotechnical properties should be considered. Each considered location is an independent scenario.

The steps below are followed for each considered HAC location (scenario):

(1) A clustering method is applied to reduce the number of blocks in each level to solve the mathematical model in a reasonable time.

(2) The clusters' precedence relationships are determined based on pit slope and mining direction.

(3) The long-term production schedule and the crusher location-relocation plan are generated simultaneously using the developed ILP formulation, and the NPV is calculated.

The scenario with the highest NPV is selected as the optimum solution.

Furthermore, the assumptions and scope of the developed optimization framework are as follows:

(a)  All the economic and technical parameters used as inputs of this model are known and deterministic

(b)  The HAC is fixed on one side of the final pit wall throughout the mine life; however, this system can be extended to deeper levels by connecting another conveyor flight

(c)  The proposed model only considers material handling cost up to the pit rim. The ex-pit facilities' location (i.e., waste dump, stockpile, and processing plant) and the material handling costs from the pit exit to the final destinations are not considered in

this model. It should be noted that these costs can be added as a fixed cost for each scenario separately.

(d)    The HAC is used to transfer both ore and waste; therefore, parallel conveyors should be installed to move materials.

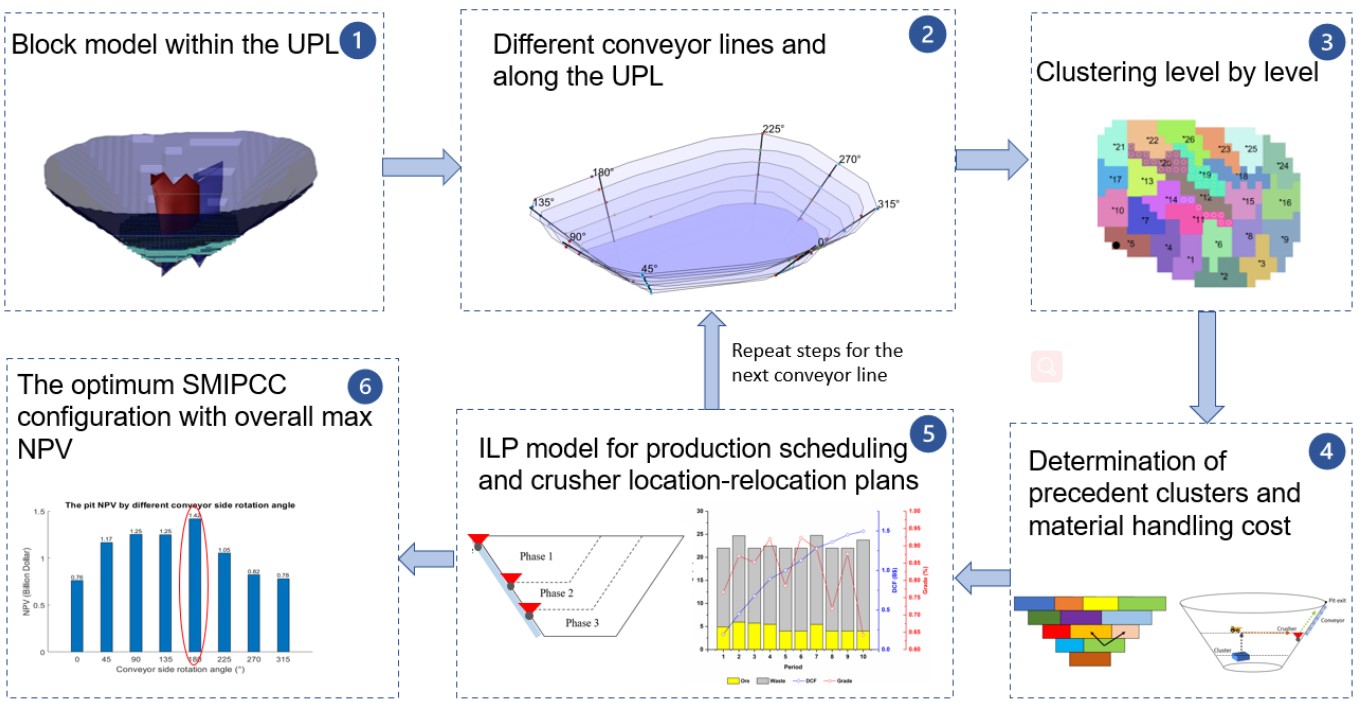

**Figure 2.** Schematic view of the methodology.

### 3.2. Determination of the HAC Location

The conveyor is located on the pit wall. A pit rotation approach is proposed to investigate the possible locations along the UPL. In this approach, each scenario's conveyor scheme is based on a series of equal interval rotation angles, making this investigation evenly distributed. Hay et al. [18] initially used this method to determine the UPL with a straight conveyor wall. This concept is used here as a tool to investigate the possible conveyor layout azimuth along the UPL.

Based on the existing UPL, a group of convex hulls is created for each level. Each convex hull is the minimum convex polygon that circumscribes all the centroids of blocks to be mined in that level, and all the interior angles are equal or less to $180°$. Then the minimum bounding box is generated from the convex hull. The bounding box is extended an additional 0.5 unit of the block width on each side to include all parts of blocks.

The bounding box's specified edge is considered the conveyor wall side (CWS), as the bold line shows in Figure 3. The tangent point between the UPL and the CWS is identified as the conveyor spot. If there is more than one point of tangency, the midpoint is considered, the red point in Figure 3. The bounding boxes are determined again at each rotation angle, and the tangent points also move around the UPL accordingly.

The bounding box clockwise rotation can be realized by rotating the x-y coordinate system, as Figure 4 shows. The rotated bounding box is generated under the new coordinate system. The rotation transformation is given by Equation (1), where the left-hand side is the new coordinates under the coordinate system with a clockwise rotation of $\alpha$ [22].

$$\begin{bmatrix} x' \\ y' \\ z \end{bmatrix} = \begin{bmatrix} \cos(\alpha) & -\sin(\alpha) & 0 \\ \sin(\alpha) & \cos(\alpha) & 0 \\ 0 & 0 & 1 \end{bmatrix} \begin{bmatrix} x \\ y \\ z \end{bmatrix} \tag{1}$$

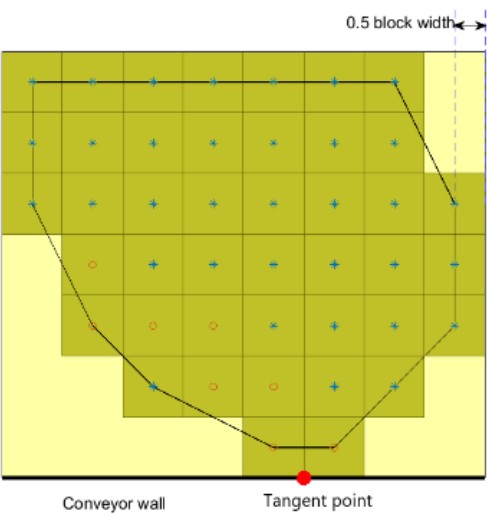

**Figure 3.** Creation of convex hull and bounding box for a specific level.

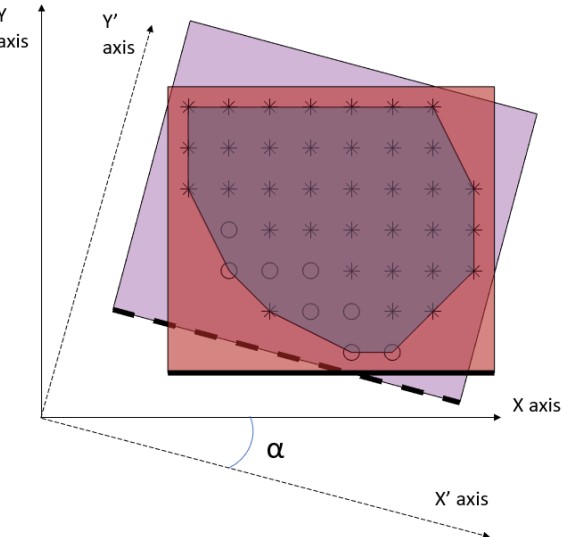

**Figure 4.** Pit level and the new bounding box with an x-y axis rotation.

It should be noted that the rotation transformation will change the original X and Y coordinates of each block; however, since the study focuses on the relative location between the conveyor and the pit, the absolute coordinates during the rotation are not a concern. Then, the coordinate system rotates clockwise by a step angle, and the associated CWS and tangent points are updated each time until it returns to the original position. Each rotation angle is a scenario for later calculation. The step angle should have a decent resolution to cover all scenarios accurately, but not too small as the computation time will increase. Besides, the pit walls' geotechnical condition plays a significant role in selecting the step angle. Considering the waste dump, stockpile, and processing plant location also affects the step angle and the pit area that is investigated. After calculating all the tangent points at all levels with the same rotation angle, the least square regression algorithm is applied to generate a straight line as the HAC layout from the pit bottom to the pit rim, as shown in Figure 5. Therefore, the conveyor line is as close as possible to each tangent point while maintaining straight line, and it is generally on the pit wall with an inclination equal to the pit slope. Moreover, two sets of HACs should be installed on the layout line for transporting ore and waste, respectively. This setting can also increase the throughput of the conveying system as the capacity of HAC is generally lower than the conventional conveyor. The capacity of each HAC may be different depending on the overall stripping ratio.

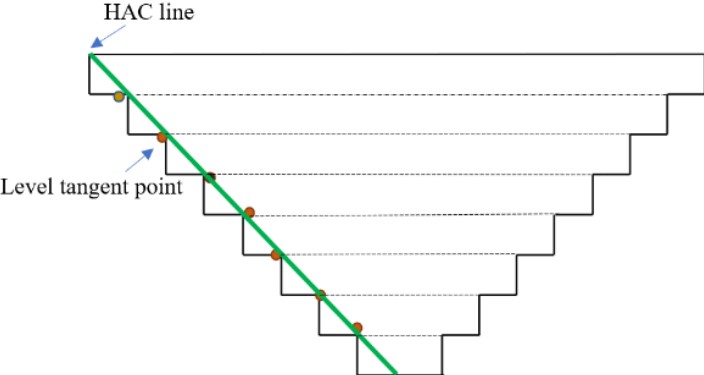

**Figure 5.** The vertical section of the pit with HAC system layout.

### 3.3. Hierarchical Agglomerative Clustering

The proposed mathematical model contains binary integer variables, which have a significant impact on the problem complexity. One of the well-known approaches is block clustering, which groups blocks with similar attributions and locations into one cluster. It can significantly reduce the number of binary integer variables and computational time. Moreover, it can generate practical mining schedules that follow a selective mining unit without scattered scheduling solutions [23]. Aggregation techniques are highly dependent on the structure of the problem and, in general, are tailored specifically for a class of problems or even for a specific instance of a problem. In this case, clusters are generated level by level, with a relatively stable size that gives both acceptable resolutions and running time. The algorithm presented by Tabesh and Askari-Nasab [24] was modified for the proposed methodology.

The similarity value between block *i* and *j* is calculated using rock type, grade, Euclidean distance, and mining direction. Blocks with more similar attributes have higher similarity values; thus, they are grouped in one cluster. The similarity between block *i* and *j* is calculated by Equation (2)

$$S_{ij} = \frac{RT^{w_{RT}}}{Dis^{w_{dis}} \times Gr^{w_{gr}} \times Dir^{w_{dir}}} \tag{2}$$

In which *Dis*, *Gr*, *Dir*, and *RT* are the normalized value of Euclidean distance, grade difference, mining direction difference, and rock type parameter between block *i* and *j*, respectively, and $w_{dis}$, $w_{gr}$, $w_{dir}$, and $w_{RT}$, in the power position, are a set of positive numbers denoting the weights of corresponding parameters. Setting a higher weight for a specific parameter can promote the clustering results to follow that characteristic. For example, increasing $w_{dis}$ can create rounder clusters while increasing $w_{gr}$ makes clusters more compliant with the grade distribution.

In the presented methodology, mining operations should be expanded from the conveyor spot to the other side of the level (Figure 6a). The initial cut of a specific level is opened at its conveyor spot, from where the mining equipment is placed in the same level, and the mining progression of that level begins. Blocks are accessed by the excavator from the conveyor spot's direction, and the material is trucked from the active mining faces towards the conveyor side (Figure 6b). In this sense, additional mining sequences should be considered; blocks closer to the conveyor location should be mined before other blocks, leaving space for the mining equipment to access the further blocks and transport the material toward the conveyor. Thus, the general mining direction at a level can be determined by the conveyor spot.

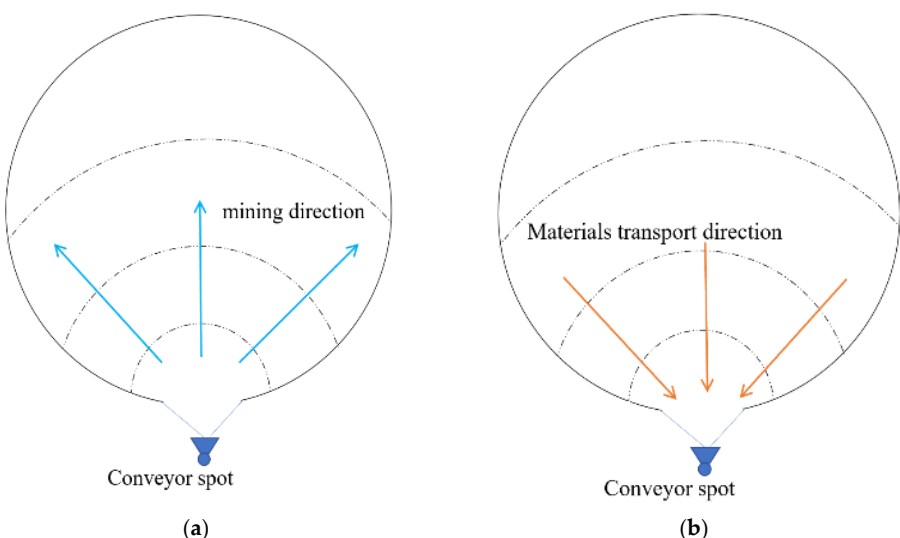

**Figure 6.** Illustration of mining and materials transport direction for a certain level: (**a**) mining direction and (**b**) material transport direction.

Blocks with the same distance to the conveyor spot are more likely to be mined in the same period; thus, they should be assigned more similarity in mining direction. In other words, the long sides of clusters should face the conveyor location.

Figure 7 shows the mining direction effects on clustering, where the clusters are shown in rectangular. Clusters with darker colors are mined earlier; those with the same color, because their distance to the conveyor is similar, they are more likely to be extracted in one period. Therefore, the mining direction difference between them should be small.

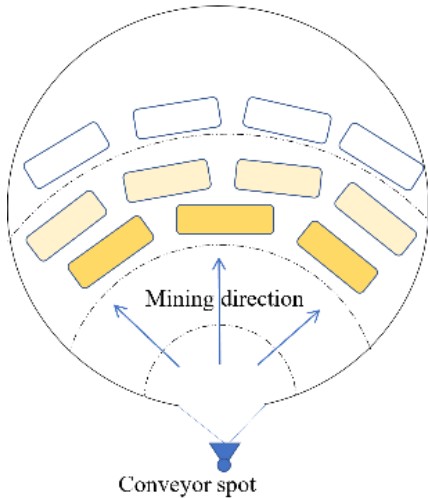

**Figure 7.** Diagram of generated clusters considering mining direction.

The mining direction difference measures the difference between two blocks' distance to the conveyor spot. The Equation (3) shows the calculation of the normalized mining direction difference $Dir_{ij}$ between block $i$ and $j$ at the same level, where $(x_o, y_o)$ is the coordinate of the conveyor spot at that level and $(x_i, y_i)$ and $(x_j, y_j)$ are the coordinate of block $i$ and $j$, respectively. The denominator is the maximum distance between the cluster centroid and conveyor spot at that level.

$$Dir_{ij} = \frac{\left| \sqrt{(x_i - x_o)^2 + (y_i - y_o)^2} - \sqrt{(x_j - x_o)^2 + (y_j - y_o)^2} \right|}{\max \{\text{distance to conveyor spot}\}} \tag{3}$$

In the modified clustering algorithm, a merge process is proposed to merge the clusters smaller than the tolerance to adjacent clusters with the same rock type. The lower size tolerance is set at 80% of the target cluster size.

Figure 8 shows an example of the modified clustering algorithm. The target cluster size is 20 blocks. The centroid of each cluster is dotted with the cluster number beside it; ore blocks are marked with white circles, and the big black dot shows the level's conveyor location.

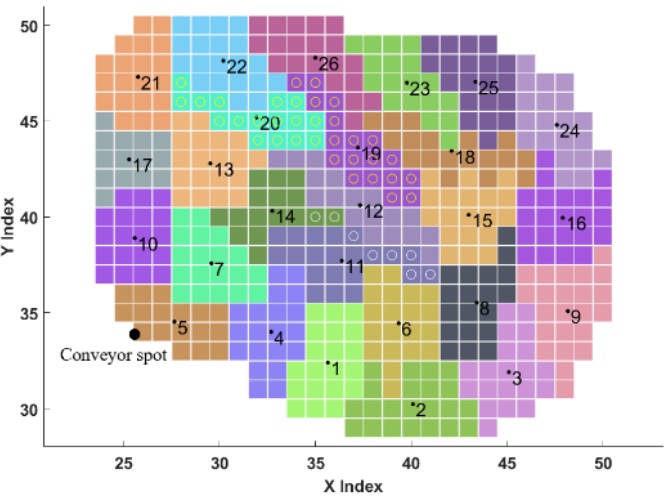

**Figure 8.** Clustering results for a level by applying the modified clustering algorithm.

### 3.4. Cluster Precedence Relationships

The precedence relationships are divided into two types (i) horizontal precedence at the same level and (ii) vertical precedence between two levels.

The horizontal precedence is a result of the mining direction. At a specific level, the precedent clusters must be extracted in advance to make the target cluster available for mining. The precedence clusters should be (i) adjacent to the target cluster and (ii) their center point should be closer than the center point of the target cluster to the conveyor spot. Figure 9 illustrates the horizontal precedence among clusters based on the mining direction. The target cluster numbered 13 has two direct precedent clusters numbered 12 and 14, where both clusters' centers are closer to the conveyor spot than the target one.

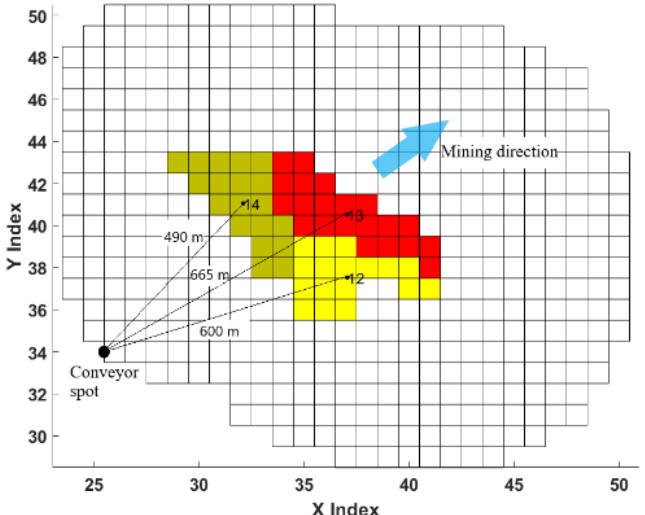

**Figure 9.** Schematic view of the direct horizontal precedence.

The vertical precedence is challenging to determine at the cluster level. To be specific, the shapes of the generated clusters are irregular. The clustering process is independent level by level, so the relative location of clusters between two adjacent levels is unspecified. The vertical precedence is determined in two steps as follows:

***Step 1:*** precedent blocks of each block within the target cluster are defined based on the classic precedence rule. The predecessors can be found from the upper-level blocks based on their coordinates or indices (see Figure 10).

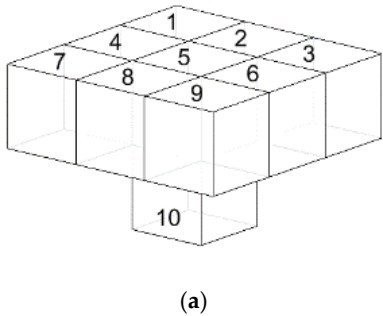

|     (a)     |     (b)     |

**Figure 10.** Nine predecessors' pattern to control pit slope: (**a**) to mine the block numbered 10, its nine predecessors at the upper level should be mine first and (**b**) the relative indices of nine predecessors in the 2D plan.

***Step 2:*** along the blocks' border, a boundary is created to envelop all the target cluster's precedent blocks. Figure 11 shows the target cluster outlined by a bold black line, and clusters 11, 12, 14, 16, and 17 are located at the upper level. The red line is the precedent blocks' boundary in the upper level. The materials from the upper level and inside the red boundary line should be extracted before the target cluster. Based on the red boundary line, each precedent cluster can be identified if any of the following criteria are satisfied:

1.  Cluster's centroid is inside the red boundary (clusters 14, 16, and 17)
2.  Cluster is partly inside the boundary, and the conveyor is closer to the cluster's centroid than to the centroid of the target cluster (clusters 11, 12, and 17)
3.  The portion of the cluster inside the red boundary is greater than 40%.

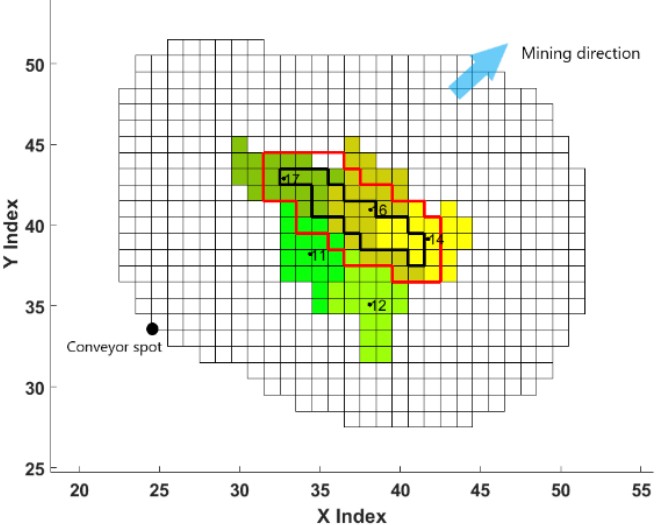

**Figure 11.** Vertical precedent clusters (clusters 11, 12, 14, 16, and 17) for the target cluster at the lower level (in bold black outline).

All these five clusters (11, 12, 14, 16, and 17) are the target cluster's vertical predecessors, and they are one level upward. The flowchart of the determination process is shown in Figure 12.

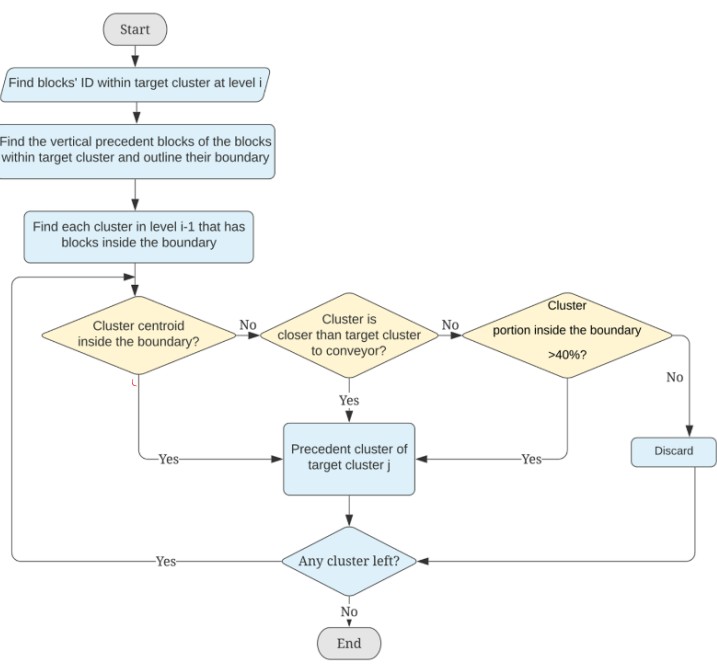

**Figure 12.** Flowchart for determining the precedent clusters.

*3.5. Material Handling Costs*

In semi-mobile IPCC systems, a small truck fleet is required to haul run-of-mine rocks from the loading point to the crusher and from where the conveyor belt system sends the crushed material to the pit exit. If the crushing station is closer to the working face, the trucking distance will be shorter, and correspondingly, the conveying portion will increase. On the other hand, conveyor belts have a cheaper operating cost than trucks. Therefore, the crusher location can determine the trucking and conveying portions of the material handling and their costs. The material handling cost per unit weight is estimated at the cluster level based on the coordinates of each cluster's centroid and the crusher location. This cost is divided into the trucking and conveyor costs: the former calculates the trucking portion from the loading point to the crushing station, and the latter measures the conveying portion from the crushing station to the pit exit.

Although the truck hauling road is a continuous ramp with curves and switchbacks, the trucking part is further divided into horizontal and vertical components for the cost calculation. Compared with hauling horizontally, trucks travel considerably longer distances and consume more energy in hauling material to a different level. The vertical difference during the trucking part should be considered separately. The components of material handling for the cost estimation are illustrated in Figure 13.

The material handling costs are estimated through the Euclidean distance of each part, Equation (4). It calculates the material handling cost per unit weight (denoted by $F_{ij}$) for a specific cluster $i$ with respect to the crusher location $j$.

$$F_{ij} = \underbrace{DH_{ij} \times CT_H}_{\text{horizontal trucking}} + \underbrace{DV_{ij}^T \times CT_V}_{\text{vertical trucking}} + \underbrace{DV_j^C \times CC}_{\text{conveying}} \tag{4}$$

where $DH_{ij}$ is the horizontal trucking distance from the cluster $i$ to the crusher location $j$; $CT_H$ is the unit horizontal trucking cost; similarly, $DV_i^T$ and $DV_j^C$ are the vertical trucking

distance from cluster $i$ to the crusher location $j$ and vertical conveying distance from crusher location $j$ to the surface, respectively; and $CT_V$ and $CC$ are their corresponding unit cost.

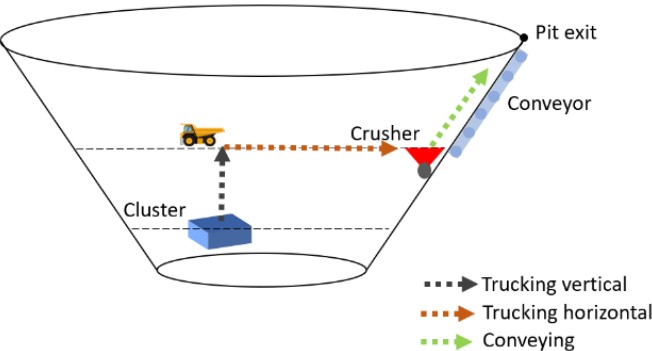

**Figure 13.** Diagram of three parts of material handling cost (showing in dash line arrows: trucking vertical, trucking horizontal, and conveying).

## 4. Integer Linear Programming (ILP) Model

Solving any mathematical model requires specified parameters that correspond to a scheduling program. This section presents an ILP model for the semi-mobile IPCC production scheduling with the fixed HAC system. ILP refers to optimization problems containing integer variables while the objective function and the constraints (other than the integer constraints) are linear. The model is formulated at the cluster level, and each cluster is considered as a mining unit. The objective is to maximize the NPV, which is the summation of discounted cluster economic values. Table 1 shows the different sets, indices, parameters, and decision variables used in the model.

**Table 1.** Overview of the sets, indices, parameters, and decision variables used in the model.

| Sets | |
|---|---|
| $\mathbb{N}$ | Set of clusters in the model |
| $\mathbb{B}_i^w$ | Set of waste blocks in cluster $i$ |
| $\mathbb{B}_i^o$ | Set of ore blocks in cluster $i$ |
| $\mathbb{P}_i$ | For each cluster $i$, there is a set of immediate predecessors that must be extracted before extraction of cluster $i$ |
| **Indices** | |
| $i \in \{1, \ldots, I\}$ | Index for clusters |
| $j \in \{1, \ldots, J\}$ | Index for pit levels |
| $t \in \{1, \ldots, T\}$ | Index for scheduling periods |
| **Parameters** | |
| $I$ | Total number of clusters |
| $J$ | Total number of levels |
| $T$ | Number of scheduling periods |
| $r$ | Discount rate |
| $CLEV_i$ | The undiscounted economic value of cluster $i$ |
| $Ton_b$ | The tonnage of block $b$ |
| $Ton_i^w$ | The total tonnage of waste in cluster $i$ |
| $Ton_i^o$ | The total tonnage of ore material in cluster $i$ |
| $g_b$ | Grade of ore block $b$ |
| $g_i^o$ | The average grade of ore material in cluster $i$ |
| $Pc_t$ | Price per unit of product sold in period $t$ |
| $s$ | Selling cost per unit of product |
| $m^w$ | Cost of mining a ton of waste |
| $m^o$ | Cost of mining a ton of ore |

**Table 1.** *Cont.*

| Parameters | |
|---|---|
| $c^o$ | Cost of processing a ton of ore |
| $R$ | The recovery rate for ore material |
| $f_{ijt}$ | Discounted transportation cost for cluster $i$ sent to crusher $j$ in period $t$ |
| $F_{ij}$ | Material handling cost per unit weight of cluster $i$ to crusher $j$ |
| $c_t$ | Discounted crusher relocation cost at period $t$ |
| $n$ | The minimum period interval for crusher relocation |
| $DH_i$ | The horizontal distance from the centroid of cluster $i$ to the crushing station $j$ |
| $DV_{ij}{}^T$ | The vertical distance from cluster $i$ to the crushing station $j$ (truck hauling) |
| $DV_j{}^C$ | The vertical distance from crushing station $j$ to the pit exit (conveying) |
| $CT_H$ | Unit truck horizontal hauling cost per ton per meter |
| $CT_V$ | Unit truck vertical hauling cost per ton per meter |
| $CC$ | Unit conveyor vertical lifting cost per ton per meter |
| $\overline{M}^t$ | Upper bound of mining capacity in period $t$ |
| $\underline{M}^t$ | Lower bound of mining capacity in period $t$ |
| $\overline{P}^t$ | Upper bound of processing capacity in period $t$ |
| $\underline{P}^t$ | Lower bound of processing capacity in period $t$ |
| $\overline{G}^t$ | Upper bound on allowable average grade of processed ore in period $t$ |
| $\underline{G}^t$ | Lower bound on allowable average grade of processed ore in period $t$ |
| **Decision variables** | |
| $x_{i,t} \in \{0,1\}$ | Binary variable equal to 1 if cluster $i$ is mined in period $t$; 0 otherwise |
| $x'_{i,t} \in \{0,1\}$ | Binary variables equal to 1 if the precedent clusters are all cleared for cluster $i$; 0 otherwise |
| $x''_{ijt} \in \{0,1\}$ | Binary variables denoting if cluster $i$ is crushed at level $j$ in period $t$ |
| $y_{j,t} \in \{0,1\}$ | Binary variables equal to 1 if the crusher is in level $j$ in period $t$; 0 otherwise |
| $z_{j,t} \in \{0,1\}$ | Binary variables equal to 1 if the crusher is relocated to level $j$ in period $t$; 0 otherwise |

### 4.1. Objective Function

The objective function in Equation (5) is to maximize the NPV while considering the material handling and crushing station relocation costs.

$$Maximize \underbrace{\sum_{t=1}^{T} \sum_{i=1}^{I} \left[ \frac{CLEV_i}{(1+r)^t} \right] \times x_{i,t}}_{\text{discounted cluster values}} - \underbrace{\sum_{t=1}^{T} \sum_{j=1}^{J} \sum_{i=1}^{I} x''_{ijt} \times f_{ijt}}_{\text{material handling cost}} - \underbrace{\sum_{t=1}^{T} \sum_{j=1}^{J} c_t \times z_{j,t}}_{\text{relocation costs}} \tag{5}$$

The objective function comprises three items. The first item is the summation of the discounted clusters' economic values (CLEV). The CLEV, defined by Equation (6), is the summation of each block's economic value within the same cluster.

$$CLEV_i = \underbrace{(Ton_i^o \times g_i^o \times R) \times (Pc_t - s)}_{\text{revenue}} - \underbrace{(Ton_i^o \times m^o + Ton_i^w \times m^w)}_{\text{mining cost}} - \underbrace{Ton_i^o \times c^o}_{\text{processing cost}} \tag{6}$$

The second item calculates the material handling costs based on the material flow decision variables $x''_{ijt}$ and the cost coefficients $f_{ijt}$. The value of $f_{ijt}$, calculated by Equation (7), is the discounted material handling costs of the whole cluster $i$ transferred via the crushing station in level $j$ at period $t$, where $F_{ij}$ is the unit weight transportation cost calculated by Equation (4).

$$f_{ijt} = F_{ij} \times (Ton_i^o + Ton_i^w) \times \frac{1}{(1+r)^t} \tag{7}$$

Although the mathematical model is built at the cluster level, the blocks defined as ore or waste within a specific cluster are considered separately in the formulations. Since ore should be processed and may have a different mining cost than waste, the cost

differences, processing capacity, and grade blending constraints are specialized for ore. This disaggregation process can increase the model's resolution at the cluster level without compromising the ILP model's computation time. Each cluster's average grade is based on the ore blocks in each cluster, calculated by Equation (8).

$$g_i^o = \frac{\sum_{b \in \mathbb{B}_i^o} Ton_b \times g_b}{\sum_{b \in \mathbb{B}_i^o} Ton_b} \tag{8}$$

*4.2. Constraints*

The following constraints are part of the problem in deriving the formulation. All the constraints are linear with binary variables.

- Mining and processing capacity

Equation (9) ensures that the total tonnage of material extracted from active clusters in each period is within an acceptable range that allows flexibility for potential operational variations. Equation (10) certifies that the amount of ore mined in each period is within the processing plant's acceptable range.

$$\underline{M}^t \leq \sum_{i=1}^{I} \left( Ton_i^w + Ton_i^o \right) \times x_{i,t} \leq \overline{M}^t \qquad \forall t \in \{1, \ldots, T\} \tag{9}$$

$$\underline{P}^t \leq \sum_{i=1}^{I} Ton_i^o \times x_{i,t} \leq \overline{P}^t \qquad \forall t \in \{1, \ldots, T\} \tag{10}$$

- Blending grade

Equations (11) and (12) force the mining system to achieve the desired grade. The average grade of the element of interest must be within the acceptable range.

$$\sum_{i=1}^{I} \left( Ton_i^o \times (\underline{G}_t - g_i) \right) \times x_{i,t} \leq 0 \qquad \forall t \in \{1, \ldots, T\} \tag{11}$$

$$\sum_{i=1}^{I} \left( Ton_i^o \times (g_i - \overline{G}_t) \right) \times x_{i,t} \leq 0 \qquad \forall t \in \{1, \ldots, T\} \tag{12}$$

- Reserves

Equation (13) is the reserve constraint, which ensures that each cluster can be at most mined once. It also gives freedom to the model to decide whether a cluster is mined or not.

$$\sum_{t=1}^{T} x_{i,t} \leq 1 \qquad \forall i \in \{1, \ldots, I\} \tag{13}$$

- Precedence

Equations (14) and (15) are the precedence constraints. These ensure that clusters can only be extracted if all its precedent clusters are removed.

$$|\mathbb{P}_i| \times x'_{i,t} - \sum_{i \in \mathbb{P}_i} \sum_{t'=1}^{t} x_{i,t'} \leq 0 \qquad \forall i \in \{1, \ldots, I\}, \ t \in \{1, \ldots, T\} \tag{14}$$

$$x_{i,t} - x'_{i,t} \leq 0 \qquad \forall i \in \{1, \ldots, I\}, \ t \in \{1, \ldots, T\} \tag{15}$$

- Material flow control

Equations (16) and (17) add the material flow decision variable $x''_{ijt}$ to the model. $x''_{ijt}$ is equal to 1 if cluster $i$ is mined in period $t$ (denoted by the production scheduling variable

$x_{i,t}$), and it is crushed at the level $j$ in the same period (denoted by the crusher location variable $y_{j,t}$).

$$x''_{ijt} \leq 0.5\,(x_{i,t} + y_{j,t}) \qquad \forall i \in \{1,\ldots,I\},\ j \in \{1,\ldots,J\},\ t \in \{1,\ldots,T\} \qquad (16)$$

$$x_{i,t} + y_{j,t} - 1.5 \leq x''_{ijt} \qquad \forall i \in \{1,\ldots,I\},\ j \in \{1,\ldots,J\},\ t \in \{1,\ldots,T\} \qquad (17)$$

- Crusher location and relocation control

Equation (18) guarantees that exactly one crushing station is available in each period. Equation (19) certifies that the crushing station can only be relocated to lower levels or remain static in any period. Equations (20)–(22) set the crushing station's relocation conditions; the relocation variable $z_{jt}$ is equal to 1 only if the crushing station moved to level $j$ from another level in period t. Equation (23) controls the minimum frequency of the crushing station's relocation, where the cluster station should stay at a specific level for at least $n$ periods before relocation. Mine planner will decide about the $n$, and it is an input parameter.

$$\sum_{j=1}^{J} y_{j,t} = 1 \qquad \forall\, t \in \{1,\ldots,T\} \qquad (18)$$

$$\sum_{j}^{L} y_{j,t-1} \geq \sum_{j}^{L} y_{j,t} \qquad \forall j \in \{1,\ldots,J\}, t \in \{2,\ldots,T\} \qquad (19)$$

$$z_{j,t} \geq y_{j,t} - y_{j,(t-1)} \qquad \forall j \in \{1,\ldots,J\}, t \in \{2,\ldots,T\} \qquad (20)$$

$$z_{j,t} \leq y_{j,t} \qquad \forall j \in \{1,\ldots,J\}, t \in \{2,\ldots,T\} \qquad (21)$$

$$z_{j,t} = y_{j,t} \qquad \forall j \in \{1,\ldots,J\}, t = 1 \qquad (22)$$

$$\sum_{t=1}^{T} y_{i,t} - n \times \sum_{t=1}^{T} z_{i,t} \geq 0 \qquad \forall j \in \{1,\ldots,J\} \qquad (23)$$

## 5. Case Study and Discussion of Results

This model was applied to a small-scale copper mine dataset with 2006 blocks and six levels, where each block is 50 m × 50 m in width and 40 m in height. The UPL was predetermined by Geovia's Whittle software. The information of ore (MZ1 and MZ2) and waste blocks are shown in Table 2. Table 3 shows each bench's size and tonnage, beginning from the top (level 1). Table 4 shows the parameters implemented in the model.

**Table 2.** Summary of rock type information in the block model.

| Rock Type | Density (ton/m$^{-3}$) | Average Grade (%) | Tonnage (Mt) | Number of Blocks |
|---|---|---|---|---|
| Mineralized Zone 1 (MZ1) | 2.10 | 1.836 | 2.94 | 14 |
| Mineralized Zone 2 (MZ2) | 2.10 | 0.822 | 44.73 | 213 |
| W (waste) | 1.80 | 0 | 320.22 | 1779 |

**Table 3.** Summary of the levels' information.

| | Number of Blocks | Number of Clusters | Total Tonnage (Mt) | Ore Tonnage (Mt) | Average Ore Grade (%) |
|---|---|---|---|---|---|
| Level 1 | 498 | 25 | 91.36 | 9.28 | 0.81 |
| Level 2 | 426 | 22 | 78.24 | 8.79 | 0.90 |
| Level 3 | 356 | 20 | 65.56 | 8.45 | 0.94 |
| Level 4 | 294 | 15 | 54.56 | 8.34 | 0.86 |
| Level 5 | 238 | 13 | 44.36 | 8.03 | 0.78 |
| Level 6 | 194 | 10 | 36.31 | 7.51 | 0.76 |

**Table 4.** Input parameters of the mathematical model.

| Category | Parameter | Quantity |
|---|---|---|
| Economic factors | Reference mining cost * ($/t) | 1.5 |
| | Ore processing cost ($/t) | 3.06 |
| | Recovery (%) | 90 |
| | Cu price ($/t) | 7,936 |
| | Selling cost ($/t) | 0 |
| | Discount rate (%) | 8 |
| Transportation costs | Horizontal trucking cost ($/km·t) | 0.2 |
| | Conveyor vertical lifting cost ($/level·t) | 0.3 |
| | Vertical trucking cost ($/level·t) | 1.2 [25] |
| | Crusher relocation cost ($/time) | 1,000,000 [17] |
| Production | Upper mining capacity (Mt/year) | 30 |
| | Lower mining capacity (Mt/year) | 25 |
| | Upper processing capacity (Mt/year) | 6 |
| | Lower processing capacity (Mt/year) | 4 |
| | Upper ore blending grade (%) | 1.0 |
| | Lower ore blending grade (%) | 0.5 |
| | Mine life (years) | 10 |
| Clustering | Maximum number of the blocks | 25 |
| | Minimum number of the blocks | 15 |
| | $w_{dis}$ | 1 |
| | $w_{gr}$ | 0.2 |
| | $w_{dir}$ | 1 |
| | $w_{RT}$ | 0.2 |
| Others | Minimum crusher relocation interval (year) | 2 |

* This reference mining cost excludes material handling cost.

In this model, eight scenarios with different conveyor side rotation were calculated, from 0° to 315° with a step size of 45°. For a specific scenario, the tangent points (conveyor spots) were generated based on the levels' convex hulls and the CWS, as the small dots displayed in Figure 14. The ordinary least square algorithm was then applied to construct a regression line to fit the conveyor spots. This line is the best approximation of the conveyor spots such that the sum of the horizontal distance from each of the conveyor spots to the line is minimum, and it is considered as the HAC layout. Figure 14 shows the eight scenarios applied in the case study and the considered HAC locations (straight lines). The clustering process was repeated, and the mathematical model was solved independently for each scenario.

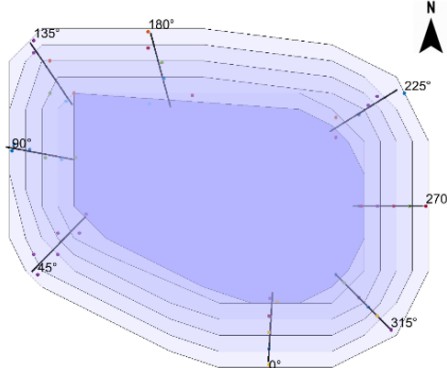

**Figure 14.** The outline of various candidate HAC lines (black straight line) based on tangent points and rotation angles.

All the steps were developed in MATLAB [26] and solved in the IBM ILOG CPLEX [27] environment. CPLEX uses a branch-and-bound scheme with an integer linear programming

solver to solve the convex ILP model, ensuring an optimal solution if the algorithm is run to completion. A gap (EPGAP) was used as an optimization termination criterion. This is a relative tolerance on the gap between the best integer objective and the best objective value among the remained nodes [27].

An Intel four-core CPU at 3.2 GHz with 6 GB RAM was used to conduct the computation, and the relative gap tolerance was set to 1%. The average CPU time for each scenario was about 280 s. Table 5 shows the results of different scenarios in detail. The total mining tonnage for each scenario fluctuates within the defined mining capacity range (25 to 30 Mt/year), while no notable relationship can be observed between its quantity to the corresponding NPV. The ore tonnage shows a more dramatic change and a more noticeable impact on the NPV under different scenarios. This is because of the different mining direction for each scenario. The average ore grade is calculated based on the weighted average grade of the ore material under a specific scenario. The optimum scenario has the maximum NPV obtained at a 180° rotation angle, with the lowest total stripping ratio. Under this scenario, the conveyor line is closer to the ore body, and less waste should be extracted before the ore body's exposure.

**Table 5.** Summary of model results for eight conveyor side rotation scenarios.

| Rotation Angle (°) | Total Tonnage (Mt) | Ore Tonnage (Mt) | Total Stripping Ratio | NPV (B$) | Average Ore Grade (%) |
|---|---|---|---|---|---|
| 0 | 287.02 | 34.48 | 7.32 | 1.023 | 0.877 |
| 45 | 258.05 | 41.33 | 5.24 | 1.204 | 0.828 |
| 90 | 258.79 | 42.43 | 5.10 | 1.232 | 0.859 |
| 135 | 286.17 | 46.17 | 5.20 | 1.432 | 0.838 |
| 180 | 277.23 | 45.75 | 5.06 | 1.468 | 0.833 |
| 225 | 292.08 | 43.32 | 5.74 | 1.306 | 0.882 |
| 270 | 278.11 | 36.91 | 6.54 | 1.118 | 0.893 |
| 315 | 282.10 | 35.14 | 7.03 | 0.932 | 0.876 |

To validate the relationship between the block value distribution and the obtained NPVs of different conveyor locations, the cumulative block economic value in the horizontal 2D plane is presented in Figure 15. The block economic value of each column within the UPL is summed up, and the linear interpolation method is applied to smooth the value distribution. From the figure, the optimum conveyor location is close to the block model's high-value area (presented in red and yellow). Because the mining operation starts from the conveyor side, the high-value area can be extracted earlier under the optimum conveyor location than the other scenarios. This time difference can impact discounted cash flow and result in different NPVs.

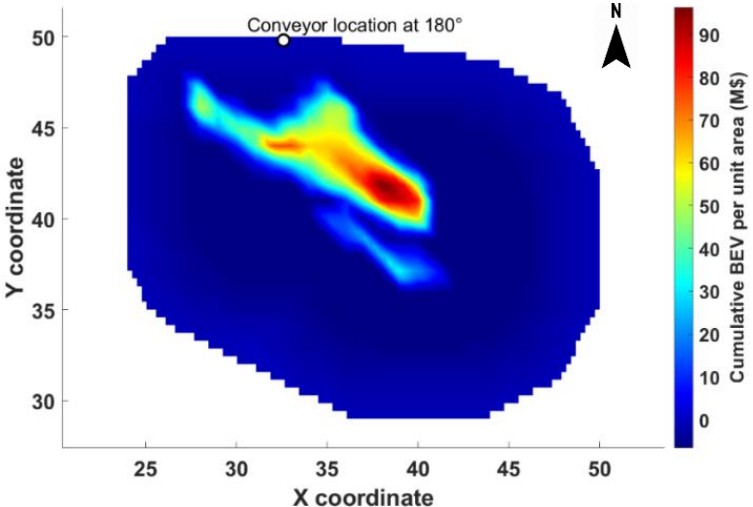

**Figure 15.** The plan view of the cumulative block economic value contour.

Figure 16 shows the production tonnage of waste and ore for the scenario with a 180° rotation angle. The blue line shows the cumulative discounted cash flow. The average ore grade has some fluctuation in different periods due to the ore grade distribution. The mining sequence of the optimal scenario generated by the ILP model is illustrated in Figure 17.

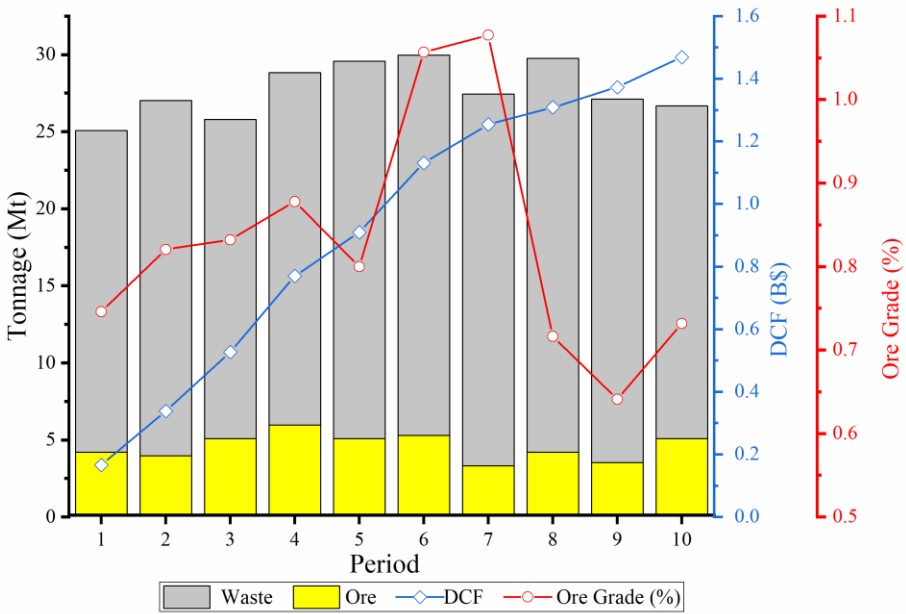

**Figure 16.** Production scheduling for the scenario with a 180° rotation angle.

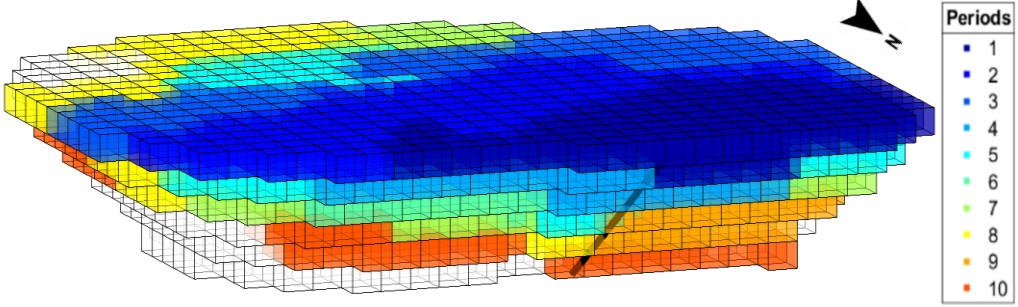

**Figure 17.** Diagram of production scheduling results for the optimum scenario.

Figure 18 shows the crusher location-relocation plan as an output of the model. The value 1 in each cell represents that the crusher is located in the associated level and period along the conveyor line. The minimum duration for the crusher stays at a certain level is set to 2 periods. The crusher is relocated three times from its initial location as the mining level goes downward. The minimum duration is an input parameter to the model and, based on the companies' strategy, can be changed. The discounted material handling cost and the total crushing station relocation cost during the mine life are $ 206.3 million and $ 2.01 million, respectively. These costs are included in the generated NPV.

Figure 19 shows the plan view of each level separately. Each colored block corresponds to a specified period that it is completely extracted. Blocks from clusters that are leftover by the end of mine life are uncolored.

| Variable values | | Period | | | | | | | | | |
|---|---|---|---|---|---|---|---|---|---|---|---|
| | | **P1** | **P2** | **P3** | **P4** | **P5** | **P6** | **P7** | **P8** | **P9** | **P10** |
| Level | **L1** | 1 | 1 | 1 | 0 | 0 | 0 | 0 | 0 | 0 | 0 |
| | **L2** | 0 | 0 | 0 | 1 | 1 | 0 | 0 | 0 | 0 | 0 |
| | **L3** | 0 | 0 | 0 | 0 | 0 | 1 | 1 | 1 | 0 | 0 |
| | **L4** | 0 | 0 | 0 | 0 | 0 | 0 | 0 | 0 | 1 | 1 |
| | **L5** | 0 | 0 | 0 | 0 | 0 | 0 | 0 | 0 | 0 | 0 |
| | **L6** | 0 | 0 | 0 | 0 | 0 | 0 | 0 | 0 | 0 | 0 |

**Figure 18.** Results of the crusher location-relocation plan.

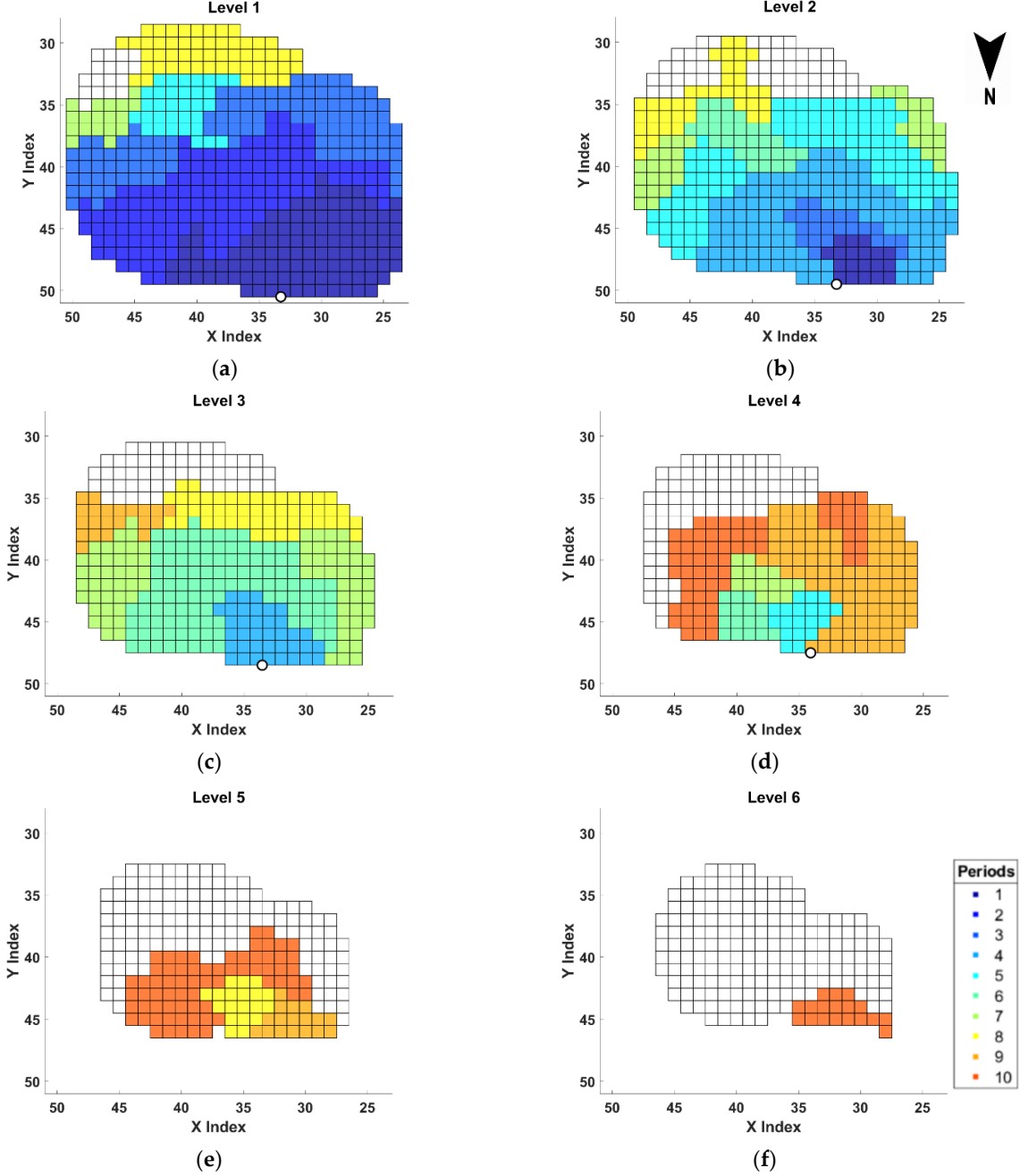

**Figure 19.** Plan view of production scheduling results from (**a**) the top level to (**f**) the bottom level with the conveyor spot (hollow circle).

The conveyor location of each level is illustrated in a small circle. It can be seen that clusters around the conveyor side are mined initially, while several blocks from the opposite side remain intact. These blocks have the least precedence and would be mine during the latter period of mine life. Thus, the profits generated by these blocks are depreciated by a higher discount factor, so that the mining cost cannot be covered. Therefore, the far side blocks are likely to be left. The total tonnage of the extracted material is 277.23 Mt, which means only 75.35% of materials inside the UPL are mined, considering the total tonnage in UPL is 367.89 Mt. In a test calculation, although the upper bound of the mining capacity is removed, some far side blocks are still not scheduled in this model. That indicates the original UPL based on the truck-and-shovel system should be updated according to the conveyor side, as materials from the opposite side to the conveyor have little contribution to the NPV. Figure 20 presents the original blocks within the UPL and the mined blocks obtained from the model's results. The blocks in dark grey are unscheduled, which should be excluded from the new UPL under the IPCC system.

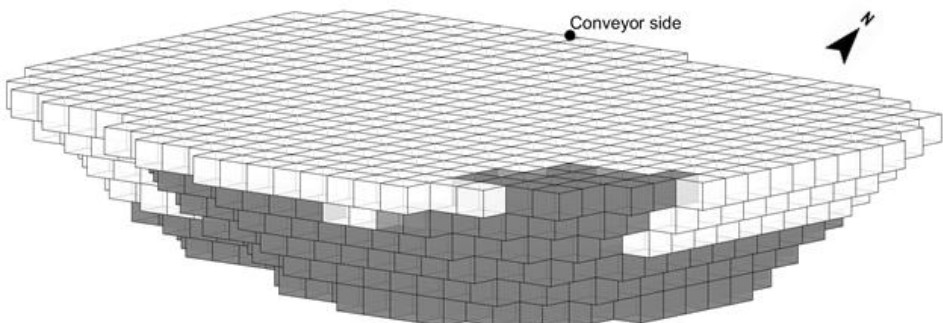

**Figure 20.** The original UPL and the mined area determined by the model (blocks in white).

## 6. Conclusions

Due to the high initial investment and the reduction in operating flexibility, IPCC systems require accurate planning before their application. This paper considered a situation that HAC is fixed in one pit side throughout the mine life and it extends to further levels when the operation goes deeper. An optimization model under semi-mobile IPCC systems was proposed, which improves the current literature on the IPCC systems' optimization planning by:

- Proposing a mathematical model for making the production scheduling and crusher location-relocation simultaneously
- Incorporating the material handling costs and crusher relocation costs in the NPV maximization
- Developing a conveyor location optimization framework by generating various conveyor lines around the UPL and finding the optimum based on the NPV comparison

The case study shows that the conveyor location can significantly impact NPV, leading to a 57.5% NPV difference between the worst and best scenarios in the considered case study. Therefore, the conveyor's layout should be designed carefully before implementing the IPCC system, especially for the conveyor line fixed through the mine life. The final pit limit should also be updated based on the obtained optimum conveyor line location and the production schedule.

**Author Contributions:** Conceptualization, D.L. and Y.P.; methodology, D.L. and Y.P.; software, D.L.; validation, D.L. and Y.P.; formal analysis, D.L. and Y.P.; investigation, D.L. and Y.P.; resources, D.L. and Y.P.; data curation, D.L. and Y.P.; writing—original draft preparation, D.L.; writing—review and editing, D.L. and Y.P.; visualization, D.L.; supervision, Y.P. All authors have read and agreed to the published version of the manuscript.

**Funding:** This research received no external funding.

**Institutional Review Board Statement:** Not applicable.

**Informed Consent Statement:** Not applicable.

**Data Availability Statement:** Not applicable.

**Conflicts of Interest:** The authors declare no conflict of interest.

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
