# Peer review of "A Framework for Open-Pit Mine Production Scheduling under Semi-Mobile In-Pit Crushing and Conveying Systems with the High-Angle Conveyor"

_mining, doi:10.3390/mining1010005_

Round 1
Reviewer 1 Report
- I propose to modify the name of article: A Framework for Open-Pit Mine Production Scheduling under Semi-Mobile In-pit Crushing and Conveying Systems with the High-Angle Conveyor.
- Please extend the detail conclusions about the found results in chapter Conclusions.
Author Response
Dear Reviewer,
We would like to thank you for the careful and thorough reading of this manuscript and the thoughtful comments and constructive suggestions, which help improve this manuscript's quality. A cover letter has been attached (See "Report Notes") based on reviewers' comments and other revisions since the last version of the manuscript.
Best regards,
The authors

Reviewer 2 Report
In this work, an integer linear programming (ILP) model is developed to investigate the long-term production scheduling in open-pit mines with a semi-mobile IPCC system and high-angle conveyor. In this context, the authors generate a series of candidate high-angle conveyor locations together with a crusher around the pit limit. The optimum or near-optimum solution's decision is based on maximising the net present value (NPV) of the open-pit mining project considering the whole cycle of material handling for all mine production period.
Combining the theoretical analysis of the model and applying it to a copper mine dataset provides a valuable tool for open-pit mines production scheduling.
The research topic is very interesting and up to date regarding open-pit mines' production scheduling, considering the related projects' whole production life-cycle. It is also important for the readers of the journal.
However, I would expect a better explanation and clarification of the research questions and a more precise presentation of the work's original contribution compared to other previous works. From a mining engineering point of view, the presentation of the case study needs to be improved. In this framework, the authors should emphasise the discussion of the results derived from applying the ILP model to the small-scale copper mine.
Additional comments and recommendations for the improvement of the manuscript:
General comments
- Compared to other previous works, the research questions and the work's original contribution need to be further clarified.
- Please discuss the advantages and disadvantages of the applied model critically
- It seems that the cost of processing a tonne of ore, which is described in the parameters of the model, is not used in the calculations. Please explain.
- Are the costs of conveying the crushed ore from the pit exit to the processing plant or conveying the waste material to the dumping area considered in selecting the optimal solution? Please explain.
- The validation of the model should be presented and justified.
- The uncertainties and limitations of the applied model need to be discussed.
- The abbreviations need to be explained in their first reference in the text.
- All the references should be checked and revised where needed. In many of them, the journals’ name is not included (e.g. 3, 4, 5, 10,13,14, 15, 21, 28).
- Introduction
General note: In this section, a more critical description is required considering the study's primary target.
[Page 2, Lines 52-54] “For the conventional …. from 29° to 44°.” A further explanation is needed.
[Page 2, Lines 68-69] “This study assumes…mine life”. The main problem of the study and the research questions need to be clarified.
2. Literature Review
General note: The current state of the research field needs to be discussed concerning the applied new mathematical framework.
3. Methodology
[Page 3-5, Lines 139-172] This part of “Methodology” maybe needs to be characterised as subsection 3.1 with a more general title and the “Determination of the HAC Location” as 3.2.
[Page 6, Line 222] “have” instead of “has” ?
[Page 7, Line 246-248] “Blocks closer to…the conveyor”. This sentence needs further explanation.
[Page 9, Line 298] Figure 10b needs to be corrected. In the first line-second box, the coordinates are “x,y+1” instead of “ “x-1, y+1”, and in the same line on the right side of the second box, they are “ x+1, y+1” instead of “x, y+1”.
- Integer Linear Programming (ILP) Model
[Page 13, Line 368] Equation 6: R instead of r.
- Case study and Discussion of Results
Critical question: Does the location of HAC affect the cost of conveying the crushed ore from the pit exit to the processing plant or conveying the waste material to the dumping area? Please comment.
[Page 15, Line 433] Does the block hight of 40 m correspond to real mining conditions?
[Page 16, Fig. 14] The location of the black straight lines, based on tangent points and rotation angles, need to be explained.
[Page 16, Table 5] (a) The differences in total tonnage and ore tonnage for each rotation angle need to be discussed, (b) The corresponding average ore grade should be added.
[Page 17, Line 480] Fig. 17 instead of Table 6
[Page 18, Fig. 17] Is the conveyor spot of the levels (e) and (f) justified considering the data of Table 6?
- Conclusions
General note: The research's original contribution must be demonstrated based on the research questions and the analysis.
Author Response
Dear Reviewer,
We would like to thank you for the careful and thorough reading of this manuscript and the thoughtful comments and constructive suggestions, which help improve this manuscript's quality. A cover letter has been attached based on the comments and other revisions since the last version of the manuscript.
Best regards,
The authors

Round 2
Reviewer 2 Report
In the revised edition, the manuscript has been significantly improved and covered my main comments